# Validation of a Lateral Flow Assay for Rapid Diagnosis of Histoplasmosis in Advanced HIV Disease, Buenos Aires, Argentina

**Mariana Andreani** [1,*], **Claudia E. Frola** [2], **Diego H. Caceres** [3,4,5], **Cristina E. Canteros** [6], **María J. Rolón** [2], **Tom Chiller** [7] and **Liliana Guelfand** [1]

1   División Laboratorio de Análisis Clínicos, Sección Microbiología, Sector Micología, Hospital Juan A. Fernández, Ciudad Autónoma de Buenos Aires C1425, Argentina
2   División Infectología, Hospital Juan A. Fernández, Ciudad Autónoma de Buenos Aires C1425, Argentina
3   Center of Expertise in Mycology Radboudumc/CWZ, 6525GA Nijmegen, The Netherlands
4   Immuno-Mycologics (IMMY), Norman, OK 73069, USA
5   Studies in Translational Microbiology and Emerging Diseases (MICROS) Research Group, School of Medicine and Health Sciences, Universidad del Rosario, Bogota 1653, Colombia
6   Departamento Micología, Instituto Nacional de Enfermedades Infecciosas (INEI), Administración Nacional de Laboratorios e Institutos de Salud (ANLIS) "Dr. Carlos G. Malbrán", Buenos Aires 1281, Argentina
7   Centers for Disease Control and Prevention (CDC), Atlanta, GA 30329, USA
*   Correspondence: mariandreani1@gmail.com

**Abstract:** Histoplasmosis is a major cause of mortality in individuals with advanced human immunodeficiency virus (HIV) disease (AHD). We evaluated in patients with AHD a lateral flow assay (LFA) developed by MiraVista® Diagnostics (MVD LFA). Histoplasmosis was defined based on the European Organization for Research and Treatment of Cancer/Invasive Fungal Infections Cooperative Group and the National Institute of Allergy and Infectious Diseases Mycoses Study Group (EORTC/MSG) case definitions. We also compared the results of this LFA with those obtained using a commercial enzyme immunoassay (EIA) developed by IMMY, Clarus *Histoplasma* GM EIA, IMMY (HGM EIA). A retrospective observational study was conducted at Hospital Juan A. Fernández, located in Buenos Aires, Argentina. The study included 48 urine specimens from patients aged >18 years with AHD. Urine specimens included 17 patients with disseminated histoplasmosis and 31 specimens from patients without evidence of histoplasmosis. Specimens were tested using the MVD LFA and the HGM EIA. The MVD LFA and the HGM EIA had similar analytical performance, with a sensitivity of 94%, specificity of 100%, positive predictive value of 100%, negative predictive value of 97%, and an accuracy of 98%. Comparison of the MVD LFA with the HGM EIA demonstrated a Kappa agreement index of 0.906. The LFA evaluated in this study had high analytical performance; it provided rapid diagnosis of histoplasmosis with minimal requirements for laboratory training, equipment, and laboratory infrastructure.

**Keywords:** histoplasmosis; HIV; rapid diagnosis; lateral flow assay

## 1. Introduction

Histoplasmosis is a systemic mycosis, acquired by inhalation of *Histoplasma* microconidia. The disease is widely dispersed, with cases reported in over 60 countries, with a particularly high incidence in the American continent [1]. People living with HIV with low CD4+ T-lymphocyte (CD4 + TL) counts are at a higher risk for developing disseminated disease [2]. This clinical form is associated with high mortality if diagnosis and antifungal therapy are delayed [1–3]. In Argentina, histoplasmosis is the second most common fungal opportunistic infection in people with advanced HIV disease (AHD) [4,5]. Prevalence of histoplasmosis in these patients ranges from 5.3% to 10% with high prevalence reported in medical facilities where highly accurate tests, such as antigen detection, are available [6].

Diagnosis of histoplasmosis is generally based on correlation of clinical, radiographic, and histopathologic evidence of disease, and laboratory tests for *Histoplasma* [7]. Reference laboratory methods to confirm the diagnosis of histoplasmosis are based on culture and microscopic observation of the fungus; however, these may delay diagnosis because of culture timelines or false negative results due to the low sensitivity of these methods [8,9].

Non-culture-based tests include antibody detection, with low sensitivity in individuals with advanced HIV infection, and in-house molecular tests, with high analytical performance but complex to implement. Antigen assays are useful tools for rapid diagnosis of the disease in patients with AHD. These assays use non-invasive specimens, such as urine [10]. Enzyme immunoassay (EIA) antigen tests have a high analytical performance, with sensitivity and specificity values over 90%, but they cross-react with other endemic fungi, such as *Paracoccidioides brasiliensis*, *Blastomyces dermatitidis*, *Talaromyces marneffei*, and less commonly, *Coccidioides* spp. [7–9,11–14]. The main limitations of these assays are the need for laboratory equipment and trained laboratory personnel to perform this testing.

MiraVista® Diagnostics (MVD) (Indianapolis, IN, USA) has recently developed a lateral flow assay (LFA), which uses polyclonal antibodies that recognize *Histoplasma* antigen in urine specimens. Recent studies have reported both EIA and LFA have comparable sensitivity and specificity, greater than 90% and have highlighted the main advantages of the LFA: it is easily implemented with minimal lab requirements, including only basic training, and it provides reliable results in less than an hour [11,15].

The goal of this study was to evaluate the analytical performance of the LFA developed by MVD for the rapid diagnosis of disseminated histoplasmosis in patients with AHD, and to correlate its results with those of the Clarus *Histoplasma* GM EIA, IMMY (HGM EIA, Norman, OK, USA), previously validated and implemented at Hospital Juan A. Fernández in Buenos Aires, Argentina.

## 2. Materials and Methods

### 2.1. Study Design and Participants

A retrospective observational study was conducted at Hospital Juan A. Fernández, Buenos Aires, Argentina. Sample size calculation was not performed; we enrolled in this study all patients with clinical suspicion of histoplasmosis attended at the Hospital Juan A. Fernández in July 2019. Urine specimens were stored at −70 °C, and we tested patients ≥18 years of age with AHD, defined as CD4 + TL counts ≤100 cell/mm$^3$ and/or clinical stage 3–4 based on World Health Organization [WHO] guidelines. Those patients who did not meet the inclusion criteria were not analyzed.

A proven case of histoplasmosis was defined following the European Organization for Research and Treatment of Cancer/Invasive Fungal Infections Cooperative Group and the National Institute of Allergy and Infectious Diseases Mycoses Study Group (EORTC/MSG) [16]. Histoplasmosis was defined by the evidence of fungal isolation, or by the presence of *Histoplasma* demonstrated microscopically using special stains. The criteria for the diagnosis of probable histoplasmosis included: evidence of environmental exposure to the fungus, compatible clinical findings, and either the presence of *Histoplasma* antigen in urine, or the presence of anti-*Histoplasma* antibodies [16].

In this study, we tested 48 urine specimens, including 17 from patients with culture-proven histoplasmosis and 31 from patients without histoplasmosis. Of the 31 patients without histoplasmosis, 18 had AHD without confirmation of other opportunistic infections, seven had mycobacterial infections, three had disseminated cryptococcosis, two had *Pneumocystis jirovecii* pneumonia, and one was diagnosed with Kaposi's sarcoma.

Both antigen detection assays were performed according to the manufacturer's instructions. In the case of the MVD LFA, a positive result involved the appearance of two lines, the test line and the control line. A negative result was considered when the control line only was present. Tests lacking a control line, independent of the test line, were considered invalid.

*Histoplasma* antigen detection by EIA was done using a commercial kit, Clarus *Histoplasma* GM EIA (HGM EIA. IMMY, Norman, OK, USA). EIA testing was performed using an automatic immunoassay analyzer (Evolis twin plus, Bio-Rad, Hercules, CA, USA).

### 2.2. Statistical Analysis

The analytical performance, sensitivity, specificity, accuracy, positive and negative predictive values, and their respective 95% confidence intervals (95% CI) were calculated using 2 × 2 tables. Agreement between MVD LFA and HGM EIA was determined using the Kappa coefficient [17].

### 2.3. Ethical Considerations

The urine specimens tested in this study were collected as part of a previous study for the validation of novel diagnostic assays for histoplasmosis. This study was conducted after approval of the Ethics Committee of Hospital Juan A. Fernández, Buenos Aires, Argentina (approval numbers: CEIHF 201329 and 201808).

## 3. Results

A total of 48 urine specimens from adults with AHD were tested. Most patients were men (n = 35; 73%), with a median age of 40 years (interquartile range [IQR]: 32–49) and with a median CD4+ count of 31 cell/mm$^3$ (IQR: 20–70) (Table 1).

**Table 1.** Demographic characteristics of patients included in the study.

| Variable | Total | With Histoplasmosis | Without Histoplasmosis |
|---|---|---|---|
| Number of patients | 48 | 17 | 31 |
| Male | 35 (73%) | 12 (71%) | 22 (71%) |
| Female | 12 (25%) | 4 (24%) | 9 (29%) |
| Transgender women | 1 (2%) | 1 (6%) | 0 |
| Age, median years (IQR) | 40 (32–49) | 37 (35–44) | 41 (31–49) |
| CD4 cell counts, median cell/mm$^3$ (IQR) | 31 (20–70) | 24 (14–50) | 56 (24–73) |

The MVD LFA had a sensitivity of 94% (95% CI: 83–94), identifying 16 out of the total 17 urine specimens from patients with histoplasmosis. We observed two false negative results, one using MVD LFA, and one using HGM EIA. The false negative result using the MVD LFA, was a specimen from a patient who had proven histoplasmosis, with isolation of the fungus by blood culture, positive antigen in urine using the HGM EIA (3.1 ng/mL), and positive *Histoplasma* RT-PCR in whole blood. The false negative result using the HGM EIA was from a patient with proven histoplasmosis (positive blood culture), positive for anti-*Histoplasma* antibodies by immunodiffusion, and positive MDV LFA (with presence of a weak test line); in addition, this patient presented a positive *Histoplasma* RT-PCR in whole blood.

Specificity of MVD LFA and HGM EIA was 100% (95% CI: 100–100), both assays presented a positive predictive value of 100% (95% CI: 100–100), negative predictive value of 97% (95% CI: 91–100), and an accuracy of 98% (95% CI: 90–100) (Tables 2 and 3). There was an agreement of 0.906 between the MVD LFA and HGM EIA (95% CI: 0.78–1.0).

**Table 2.** Results of urine specimens tested with the MVD LFA and the HGM EIA.

| | MVD LFA + | MVD LFA − | EIA + | EIA − |
|---|---|---|---|---|
| With histoplasmosis | 16 | 1 | 16 | 1 |
| Without histoplasmosis | 0 | 31 | 0 | 31 |

MVD LFA: MiraVista Diagnostics® lateral flow assay; HGM EIA: Clarus *Histoplasma* GM EIA; PPV: positive predictive value; NPV: negative predictive value.

**Table 3.** Urine testing performance of the MVD LFA and the HGM EIA.

| MVD LFA and EIA | % (95% CI) |
| --- | --- |
| Sensitivity | 94 (83–94) |
| Specificity | 100 (100–100) |
| PPV | 100 (100–100) |
| NPV | 97 (91–100) |
| Accuracy | 98 (90–100) |

MVD LFA: MiraVista Diagnostics® lateral flow assay; HGM EIA: Clarus *Histoplasma* GM EIA; PPV: positive predictive value; NPV: negative predictive value.

## 4. Discussion

The MVD LFA presented high analytical performance for the detection of *Histoplasma* antigen in urine. The results obtained in this study were similar to other validation studies conducted in Mexico and Colombia [11,15]. It should be emphasized that all these studies used urine samples from patients with advanced HIV disease, having a CD4+ cell count under 100 cell/mm$^3$ and proven histoplasmosis. Sensitivity values reported were 90% (95% CI: 83–95) in Mexico, and 96% (95% CI: 80–100) in Colombia [11,15]. In addition to rapid results and high accuracy, another advantage of *Histoplasma* Ag assays is the availability of commercially kits; this facilitates the incorporating of this type of testing into clinical laboratories. Currently, is it well known that in locations where these Ag assays have been implemented, a significantly increased in the number of cases is reported had been observed, especially compared with culture and histopathology. It has been described that the rapid access to results generate shortened time of diagnosis and a reduction of mortality associated with histoplasmosis in people living with HIV [18–26].

It is important to note that each assay uses different antibodies, which could account for the slightly different results. MVD LFA uses a polyclonal antibody and HGM EIA uses a monoclonal antibody. LFA-based methodologies may have limited analytical sensitivity compared with EIA-based methods because they do not use equipment to interpret the results, making results interpretation subjective. The false negative result obtained with the MDV LFA may have been due to low antigen concentration, below the limit of detection. The HGM EIA false negative specimen was observed in a patient with positive antibody testing; it is documented that circulating antibodies can neutralize the antigen, which may interfere with EIA detection [27]. The LFA yielded a positive result in this patient, which might be explained by the use of polyclonal antibodies on this LFA that could have allowed detection of the antigen in those epitopes not neutralized by the patient's antibodies [27]. Neither the MVD LFA nor the HGM EIA displayed false positive results, confirming the high specificity of both assays.

In 2020, the WHO and the Pan American Health Organization (PAHO) released the first guidelines for the diagnosis and management of disseminated histoplasmosis in people with HIV, recommending that disseminated histoplasmosis be diagnosed by detection of circulating antigens [28]. *Histoplasma* antigen detection by LFA has several advantages compared with other diagnostics. It requires minimal laboratory infrastructure and training, and it does not require a cold chain. Making it a point of care test.

This study had one major limitation, namely, the small sample size. We note that larger numbers of well-characterized samples from patients with proven histoplasmosis are difficult to obtain. This is also the first time that an LFA has been employed for the rapid diagnosis of histoplasmosis in Argentina. Future studies should include more urine specimens from patients with AHD, as well more urine specimens from patients with other endemic mycoses present in Argentina, such as paracoccidioidomycosis and coccidioidomycosis.

In Argentina, a substantial proportion of new diagnoses of HIV are established late in the course of illness in patients with AHD; many patients are re-presenting to care with AHD. The use of rapid, accurate, simple, and non-invasive diagnostic methods, such as the LFA validated in this study, would allow the early diagnosis and treatment of



histoplasmosis. Previous experiences have shown that rapid testing reduces mortality in patients with AHD [18–26].

**Author Contributions:** Conceptualization, M.A., C.E.F. and L.G.; methodology, M.A., C.E.F. and L.G.; validation, M.A., C.E.F. and L.G.; formal analysis, M.A., C.E.F., D.H.C. and L.G.; investigation, M.A., C.E.F., C.E.C., M.J.R., T.C., D.H.C. and L.G.; resources, M.A., C.E.F., D.H.C., T.C. and L.G.; data curation, M.A., C.E.F., D.H.C. and L.G.; writing—original draft preparation, M.A., C.E.F., D.H.C. and L.G.; writing—review and editing, M.A., C.E.F., C.E.C., M.J.R., T.C., D.H.C. and L.G.; project administration, M.A., C.E.F. and L.G. All authors have read and agreed to the published version of the manuscript.

**Funding:** This research received no external funding.

**Institutional Review Board Statement:** This study was conducted after approval of the Ethics Committee of Hospital Juan A. Fernández, Buenos Aires, Argentina (approval numbers: CEIHF 201329 and 201808).

**Informed Consent Statement:** Not applicable.

**Data Availability Statement:** The data that support the findings of this study are available from the corresponding author upon reasonable request.

**Acknowledgments:** The authors want to thank Joseph Wheat, Slava Elagin, and MiraVista Diagnostics® for the provision of assay kits and Mark Mezzullo for manuscript editing.

**Conflicts of Interest:** The authors have no conflict of interest to declare. The reagents have been provided by MiraVista® Diagnostics. On 29 November 2021, Diego H. Caceres became an IMMY employee. The results and conclusions reported in this study belong to the authors and do not necessarily reflect the official views of the U.S. CDC.

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
