# Peer review of "Validation of a Lateral Flow Assay for Rapid Diagnosis of Histoplasmosis in Advanced HIV Disease, Buenos Aires, Argentina"

_2673-8007, doi:10.3390/applmicrobiol2040072_

Round 1

Author Response

Reviewer 1

This paper validated the utility of a lateral flow assay (LFA) for rapid diagnosis of histoplasmosis in patients with advanced HIV infection in comparison with a quantitative enzyme immunoassay (q EIA) by using a urine sample.

The sensitivity and specificity of LFA were compatible to that of q EIA. In consideration of less expensive and more convenient LFA, LFA has much advantage over q EIA especially in developed countries in which expensive equipment for q EIA is difficult to prepare. In this sense, this study seems valuable clinically.

One of the authors and related collaborators have already reported the similar analysis in the Journal of Fungi (2021; 7, 799). In the Fungi paper, urine samples were from the people living with HIV/AIDS(PLHIV), and in this paper, urine samples were from the people with advanced HIV disease, who had less than 100/ µL of CD4 T lymphocytes.

There are some differences in the specificity and sensitivity between two papers in both of MVD LFA and MVD EIA. I will recommend discussing this point in terms of disease progression of HIV/AIDS.

Otherwise, the significance and novelty of this paper will be much reduced.

Minor point:

We observed two false positive results, one using MVD LFA, and one using HGM EIA.

page3

âž¡

We observed two false negative results, one using MVD LFA, and one using HGM EIA

  • Thank you for your observation. We fixed it.

Reference 1

âž¡ J Fungi (Basel 2021, 7(2): 151

Reference 2

âž¡ J Fungi (Basel 2019, 6 1): 3

Reference 8

âž¡ J Fungi (Basel 2019, 5(3) 3):76

Reference 16 and 17 are redundant.

Reference 22 âž¡ Microorganisms 2021, 9(12): 2596

R. Thank you. We fixed the references

Reviewer 2 Report

In this study, the authors validated the use of lateral flow assay (LFA) in diagnosing histoplasmosis associated with HIV. A retrospective observational study was conducted, including 48 urine specimens from patients aged >18-year-old with AHD. The sample size is small, and the study is conducted in one hospital. However, the obtained results are still interesting.

I recommend the following revision to improve the manuscript.

1.       Use the full name of the abbreviation of HIV in the abstract

2.       Use the full name of the abbreviation of EORTC/MSG

3.       Use the full abbreviation of EIA

4.       Table 1: add the standard deviation for the age means and the CD4 cells count.

5.       Define the terms sensitivity and specificity

6.       Describe the sample size selection

7.       Specify the inclusion and exclusion criteria in selecting the patients

8.       The discussion section is poor and needs to discuss the other diagnostic methods with their positive and negative aspects and compare the obtained results with other similar studies.

9.       Add future applications of the obtained results

10.   References no 2,3, 19 lack the full details

11.   Correct the presentation of reference no 6

12.   References 12, 16 and 17 are the same

Author Response

Reviewer 2

In this study, the authors validated the use of lateral flow assay (LFA) in diagnosing histoplasmosis associated with HIV. A retrospective observational study was conducted, including 48 urine specimens from patients aged >18-year-old with AHD. The sample size is small, and the study is conducted in one hospital. However, the obtained results are still interesting.

I recommend the following revision to improve the manuscript.

Use the full name of the abbreviation of HIV in the abstract

  • Changed.

Use the full name of the abbreviation of EORTC/MSG

  • Added

Use the full abbreviation of EIA

  • Added

Table 1: add the standard deviation for the age means and the CD4 cells count.

  • We prefer to keep the median and the IQR because this data do not have normal distribution.

Define the terms sensitivity and specificity

  • We don’t consider this definition necessary.

Describe the sample size selection

  • Thank you, we clarified the sample size. We added the following statement. “There was not sample size calculation, we enrolled on this all patients with clinical suspicion of histoplasmosis attended at the Hospital Juan A. Fernández in July 2019.”

Specify the inclusion and exclusion criteria in selecting the patients.

  • Thank you, we added a statement describing exclusion criteria.

The discussion section is poor and needs to discuss the other diagnostic methods with their positive and negative aspects and compare the obtained results with other similar studies.

  • Thank you, we added a new statement addressing the point suggested by the reviewer.

Add future applications of the obtained results

  • Thank you, we modified the last statement for addressing the point suggested by the reviewer.

References no 2,3, 19 lack the full details

  • Thank you, we fixed it.

Correct the presentation of reference no 6

  • Thank you, we fixed it.

References 12, 16 and 17 are the same

  • Thank you, we fixed it.

Round 2

Reviewer 2 Report

The authors addressed the comments appropriately